# Relationships between Vitamin D and Selected Cytokines and Hemogram Parameters in Professional Football Players—Pilot Study

**DOI:** 10.3390/ijerph18137124

**Published:** 2021-07-02

**Authors:** Anna Książek, Aleksandra Zagrodna, Anna Bohdanowicz-Pawlak, Felicja Lwow, Małgorzata Słowińska-Lisowska

**Affiliations:** 1Department of the Biological and Medical Basis of Sport, University School of Physical Education in Wrocław, Al. Paderewskiego 35, 51-617 Wrocław, Poland; aleksandra.zagrodna@awf.wroc.pl (A.Z.); malgorzata.slowinska-lisowska@awf.wroc.pl (M.S.-L.); 2Department of Endocrinology, Diabetology and Isotope Therapy, Wrocław Medical University, ul. Pasteur 4, 50-367 Wrocław, Poland; anna.bohdanowicz-pawlak@umed.wroc.pl; 3Faculty of Physiotherapy, University School of Physical Education in Wrocław, Al. Paderewskiego 35, 51-617 Wrocław, Poland; felicitas1@wp.pl

**Keywords:** 25(OH)D, pro-inflammatory cytokines, athletes, competitive period

## Abstract

Vitamin D affects both innate and adaptive immunity. Most of the effects of vitamin D on innate immunity are anti-inflammatory. In monocytes/macrophages, vitamin D suppresses the production of the inflammatory cytokines TNF-alpha, IL-1beta, IL-6, and IL-8. Therefore, the aim of our study was to investigate the relationship between 25(OH)D concentration and selected cytokines—IL-6, TNF-α, and IL-1β, which are hemogram parameters for professional football players. We enrolled 41 Polish premier league soccer players. The mean age, career duration, and VO_2max_ were, respectively: 22.7 ± 5.3 years, 14.7 ± 4.5 years, and 55.8 ± 4.0 mL/kg/min. Serum levels of 25(OH)D were measured by electrochemiluminescence (ECLIA) using the Elecsys system (Roche, Switzerland). Serum levels of IL-6, IL-1β, and TNF-α were measured by ELISA (R&D Systems, Minneapolis). Blood count with smear was measured on a Sysmex XT-4000i analyzer (Sysmex Corporation, Japan). Our study showed decreased serum 25(OH)D levels in 78% of the professional players. We found a significant negative correlation between 25(OH)D levels and TNF-α and LYMPH (%). The results also demonstrated a statistically significant positive correlation between vitamin D levels and NEUTH (%), NEUTH (tys/µL), and EOS (tys/µL). Based on the results of our study, we concluded that football players from Poland are not protected against vitamin D insufficiency in winter months. Moreover, vitamin D deficiency may be associated with an increased pro-inflammatory risk in well-trained athletes.

## 1. Introduction

The effect of vitamin D on calcium-phosphate balance is well established, whereas its effects on the immune system and immune processes are still being researched [1,2], as are the associations with the incidence of chronic diseases, including metabolic or neoplastic diseases. Population-based studies have revealed serum vitamin D deficiency even among young and healthy individuals, thus highlighting an important global public health problem [3]. Recent scientific reports also refer to increased immunity and decreased susceptibility to COVID-19 through vitamin D supplementation [4].

Vitamin D has been shown to stimulate the synthesis of the antimicrobial proteins cathelicidin and β-defensin by macrophages in natural killer (NK) cells [5,6,7]. Cathelicidin, along with vitamin D, increases the phagocytic properties of monocytes and macrophages [8,9,10]. In addition to its antimicrobial effects, vitamin D exhibits multidirectional anti-inflammatory properties by modulating T cell function, stimulating the differentiation of regulatory T cells, and possibly inhibiting the proliferation of particular types of T cells [11].

Many studies have shown that under the influence of vitamin D, there is a reduction in Th cell population, a decrease in the density of the cytokines IL-2, IL-9, TNF-α, IFN-γ, and IL-22, and an increase in the density of Th2 cells and the anti-inflammatory cytokines IL-3, IL-4, IL-5, and IL-10 [2,12,13,14]. It is also suggested that vitamin D has an inhibitory effect on the production of proinflammatory interleukins, including IL-1 and IL-6 [15,16,17,18,19]. Vitamin D has an inhibitory effect on Th1 cells through the suppression of Il-12 secretion by dendritic cells. This has the effect of enhancing the action of Il-10 and Treg cells, which in turn contributes to the conversion of Th1 cells into Th2 cells. This effect of vitamin D causes an increase in the number of Th2 cells that produce anti-inflammatory cytokines and a decrease in the number of Th1 cells responsible for secreting pro-inflammatory cytokines [20].

Furthermore, hemogram-derived inflammatory markers such as mean platelet volume (MPV), range of red blood cell distribution (RDW), neutrophil to lymphocyte ratio (NLR), and platelet to lymphocyte ratio (PLR) have attracted much attention in recent times. It has been shown that vitamin D deficiency can cause an increase in the mean platelet count and platelet volume, even in healthy subjects, hence, leading to the hypothesis that vitamin D deficiency may affect the parameters of the hemogram [21]. The PLR and NLR have been used to determine inflammation in different types of malignancies, infectious diseases, metabolic syndrome, cardiovascular disease, and other inflammatory diseases [22,23,24,25]. In the case of cardiovascular disease, inflammation and vitamin D deficiency may be associated with an increased risk of left ventricular concentric remodeling [26], arterial stiffness [27], endothelial dysfunction [28] and have emerged as an independent risk factor for all-cause and cardiovascular mortality [29]. The negative effects of vitamin D deficiency on the cardiovascular system may also adversely affect athletic performance. It should be noted that PLR and NLR are not widely used to determine the association of inflammation and vitamin D deficiency.

Similarly, the correlation between serum 25(OH)D levels and circulating eosinophils is not strongly established; however, a few studies have noted that lower levels of vitamin D are associated with an increased blood eosinophil count [30,31].

Athletes seem to be prone to vitamin D deficiency [32,33,34,35,36,37,38]. Unfortunately, even outdoor training (such as by football players) is not fully protective against disturbances in this regard [23,24,25,26,27,28,29,30,31,32,33,34,35,36]. On the other hand, inflammation induced by training and game play might be related to low levels of vitamin D in athletes [39]. Therefore, the aim of our study was to investigate whether there is a relationship between 25(OH)D concentration and selected cytokines—IL-6, TNF-α, and IL-1β—in the hemogram parameters of football players.

## 2. Materials and Methods

### 2.1. Participants

A total of 41 Polish premier league football players were included in the study. Participants’ characteristics are shown in Table 1. The study was conducted during a winter season in Wroclaw, Poland, which is situated at the latitude of 51°10′ N. The uniforms covered 80% of the competitors’ bodies. All the players were in the competitive period and had similar training loads. None of the subjects used any food supplements containing vitamin D or calcium.

### 2.2. Measurements

Height was measured with an anthropometer and body mass was measured with an electronic scale. Body composition (body fat, fat free body mass (FFM), total body water (TBW), muscle mass) was determined using the single frequency bioelectric impedance analyzer (BIA, 50-kHz) manufactured by Akern Bioresearch (Italy).

Aerobic performance (VO_2max_/VO_2peak_) was assessed using a 20 m multistage shuttle running field test [40].

This method most closely reflects the football effort due to the environment in which physical effort is exerted [41].

### 2.3. Blood Testing

Blood samples were collected during winter season (December-January). Blood sampling was carried out at 8 am after a 12 h fast and a 24 h period without training. Serum was separated and stored at −70 °C. Serum levels of 25-hydroxycholecalciferol were measured by electrochemiluminescence (ECLIA) using the Elecsys system (Roche, Switzerland). Serum levels of IL-6, IL-1β, and TNF-α were measured by ELISA (R&D Systems, Minneapolis). For 25(OH)D, the intra- and interassay coefficients of variation (CVs) were 5.6% and 8.0%, respectively, and the limit of detection was 10 nmol/l (4 ng/mL). The respective values for IL-6 were 2.6%, 4.5%, and 0.70 pg/mL, those for TNF-α were 4.7%, 5.8% and 1.6 pg/mL, and those for IL-1β were 4.8%, 5.6% and <1 pg/mL.

Blood counts with smear were measured on a Sysmex XT-4000i analyzer (Sysmex Corporation, Japan) (Table 2). 

In addition, inflammatory markers derived from hemograms, i.e., platelet/lymphocyte ratio (thousand/µL) (PLR) and neutrophil/lymphocyte ratio (thousand/µL) (NLR), were also calculated [21].

The mean values of evaluated parameters—WBC, NEUTH (%), LYMPH (%), EOS (%), NEUTH (thousand/µL), and EOS (thousand/µL)—in the study group were within the normal ranges, and the instances of slightly higher results could be considered clinically insignificant.

### 2.4. Statistical Analysis

Statistical analyses were performed using PQStat for Windows (version 1.4.4.126) (PQStat Software, Poznań, Poland). Continuous variables were first analyzed for normal distribution using the Kolmogorov–Smirnoff test with the Lilliefors correction. Differences in cytokine and hemogram parameter values between the two serum 25(OH)D level ranges, i.e., <20 ng/dL vs. ≥20 ng/dL were calculated by t-Student’s test or Mann–Whitney rank sum test. The relationship between 25(OH)D levels and cytokine levels was analyzed by estimating the Spearman rank correlation coefficient. Data were presented as means ± SD, with *p* < 0.05 being indicative of statistical significance.

## 3. Results

The results of our study are presented in Table 3, Table 4 and Table 5.

The mean serum levels of cytokines were as follows: IL-6—0.84 ± 0.7 pg/mL, IL-1β—2.23 ± 2.04 pg/mL, and TNF-α—1.46 ± 0.26 pg/mL. The mean serum 25(OH)D level was 16.78 ± 8.21 ng/mL. In line with the latest guidelines, the normal ranges for serum 25(OH)D levels are defined as 30–50 ng/mL (75–125 nmol/L) or 40–60 ng/mL (100–150 nmol/L). Vitamin D insufficiency is defined as a serum level of 20–30 ng/mL (50–75 nmol/L) and vitamin D deficiency as a serum level below 20 ng/mL (<50 nmol/L) [39]. We found that 12.2% (*n* = 5) of the players had normal levels of 25(OH)D, 9.8% (*n* = 4) had 25(OH)D insufficiency, and 78% (*n* = 32) had 25(OH)D deficiency. In our paper, we divided football players into two ranges, i.e., 25(OH)D <20 ng/mL vs. 25(OH)D ≥20 ng/mL [42,43].

Table 3 show differences between the study groups (differing in 25(OH)D levels) in terms of cytokine levels and hemogram parameters. The football players with 25(OH)D levels of <20 ng/mL had statistically higher LYMPH (%) (*p* = 0.0245) and lower EOS (tys/µL) (*p* = 0.028) and NLR (*p* = 0.035) compared to athletes with higher 25(OH)D levels. The remaining results for cytokines and hemogram parameters were not statistically different between the study groups.

Table 4 provides the Spearman correlation coefficient values for the relationship between 25(OH)D levels and the study variables. A statistically negative correlation was found between 25(OH)D concentration and TNF-α in the study group (*p* = 0.004). No correlations were revealed between 25(OH)D levels and IL-6 and IL-1β.

The values of the Spearman coefficient of correlation between 25(OH)D levels and hemogram parameters in studied subjects are given in Table 5. According to our results, there were statistically significant positive correlations between 25(OH)D and NEUTH (%) (*p* = 0.02), NEUTH (tys/µL) (*p* = 0.02), and EOS (tys/µL) (*p* = 0.049). There were also statistically significant negative associations between 25(OH)D levels and LYMPH (%) (*p* = 0.02). In the case of the remaining parameters, no statistically significant correlation was shown.

## 4. Discussion

Our data confirmed a relatively high prevalence of low serum concentrations of vitamin D in professional male athletes [32,33,34,35,36,37,38]. According to various observations, vitamin D deficiency can occur in up to 50–80% of the population [42].

The footballers in our study were evaluated during winter season. They all trained outdoors 2 h a day (exposing only faces and hands). In our location, sunlight exposure does not provide adequate vitamin D synthesis in autumn and winter [44]. Contrary to expectations, the mean concentration of 25(OH)D in athletes was lower than that found in non-athletes [45]. However, some findings did not support such a relationship, pointing to the balanced diet of athletes as a protective factor against vitamin D deficiency [46]. Undertaking outdoor training seems to be associated with higher concentrations of 25(OH)D as compared with exercising indoors [38]. Lower vitamin D levels are more commonly observed in athletes who train indoors [34,38,47]. Many studies performed in European football players, particularly during the winter season, showed serum 25(OH)D levels below the recommended range [48,49].

Recent publications indicate a significant relationship between physical exercise, immune system function (concentrations of pro-inflammatory and anti-inflammatory cytokines), and the status of vitamin D supply. These findings may be particularly important for professional athletes, in whom vitamin D deficiency may additionally increase the risk of inflammatory processes resulting from intense physical training and competition [50,51]. Athletes practicing high-performance sports are therefore a special group of individuals exposed to factors that result in high concentrations of pro-inflammatory cytokines. Main et al. [52] showed an increase in TNF-α and Il-6 as exercise continued, which was associated with muscle damage and the appearance of an inflammatory reaction.

Increases in CRP, Il-6, and hepcidin after exercise have also been observed in subjects in sport training [53,54]. On the other hand, in another study, Santos et al. [55] reported an increase in Il-6, Il-1ra, Il-8, and Il-10 (but not TNF-α or Il-1β) levels in athletes after they completed a marathon. In contrast, Wadley et al. [56] also observed exercise-dependent changes in interleukin concentrations in subjects who were not in training.

Physical activity increases cortisol and epinephrine blood levels, releases IL-6 from active muscles, and increases Il-10 production [57]. Despite its anti-inflammatory effects, cortisol leads to muscle damage, resulting in the appearance of pro-inflammatory cytokines in the blood [55]. On the other hand, Toft et al. [58] demonstrated an association with an increase in Il-6 concentration during exercise; however, the increase in muscle damage markers (creatinine kinase and myoglobin) was disproportionate to the minor increase in Il-6 concentration. Thus, the increase in Il-6 levels after exercise is probably not solely due to muscle cell damage [59].

Elevated TNF-α, INF-γ, IL-1β, and IL-2 levels have been observed in patients with vitamin D deficiency [60]. On the other hand, He et al. [61], demonstrated an association of decreased production of the cytokines TNF-α, IL-6, INF-γ, Il-2, and Il-10 with high plasma vitamin D concentrations (values of 1000 and 10 000 pmol/L) in athletes who practice endurance sports. This study also demonstrated the effect of vitamin D deficiency on increased circulating INF-γ and Il-10 interleukins. The 2013 study by He et al. [62], as did our pilot study, involved athletes training in winter (longitude 53°N). This study additionally demonstrated an association of a higher incidence of upper respiratory tract inflammations in athletes with vitamin D deficiency [63]. Vitamin D deficiency and the associated increase in pro-inflammatory cytokines may decrease overall immunity in athletes. In our study, we confirmed the significant negative association of serum vitamin D levels with TNF-α and LYMPH values.

Studies evaluating the correlation between serum vitamin D levels and cytokine levels are inconclusive.

In a 3-month prospective study, Yusupov et al. [61] evaluated the effect of vitamin D supplementation on the levels of cytokines (Il-2, 4, 5, 6, 8, 10, and 13) and GM-CSF, INF-gamma, and TNF-α. They showed no statistically significant changes on cytokine levels in a healthy adult group [61]. Other researchers, when studying 95 healthy adults in Japan, found no association of vitamin D levels with IL-6 or interferon-gamma. Instead, they showed an association with Il-17, regardless of physical activity level. In their group, as in ours, vitamin D deficiency was observed in more than 50% of the subjects [2]. In view of these data, it seems reasonable to monitor vitamin D concentrations in physically active individuals, including competitive athletes.

Peterson et al. [64] measured TNF-α levels in healthy women in relation to serum vitamin D levels. They showed a correlation between increasing vitamin D levels and decreasing TNF-α levels [64], which confirms our results. TNF-α is produced by macrophages, monocytes, T cells, adipocytes, and fibroblasts. Vitamin D supplementation in patients with myocardial damage has been shown to significantly reduce levels of this pro-inflammatory cytokine [65]. It seems that the negative effect of exercise on the elevation of pro-inflammatory cytokines could be minimized by ensuring adequate serum vitamin D concentrations. However, determining accurate vitamin D levels with such effects in athletes requires further research.

The analysis of the association between vitamin D levels and morphotic blood elements showed a significant negative correlation only with the percentage of lymphocytes and a significant positive correlation with the number and percentage of neutrophils; it is difficult to say whether such a correlation may indicate a higher risk of infection in vitamin D deficient subjects. No association of vitamin D with NLR or PLR was shown. The NLR index in the vitamin D deficiency group was even significantly lower than in the group with normal supply. The association of those inflammatory indices with vitamin D deficiency was pointed out by Akbas et al. [66], who reported significantly higher NLR and PLR indices in a group of 3326 subjects with vitamin D deficiency compared to 794 subjects with normal supply. The cited authors indicated that PLR index may be an independent predictor of vitamin D deficiency. In addition, Erkus et al. [22] showed that NLR and MPV can be markers of inflammation associated with vitamin D deficiency (the authors studied 85 subjects, out of whom 45 presented with vitamin D deficiency). According to that study, NLR values of >1.69 had 76% sensitivity and 55% specificity for vitamin D deficiency. The results of our study do not support these findings, probably primarily due to the small size of the study group. Taking into account the literature data, however, it seems that it is worth examining these indices, particularly because the tests are inexpensive and widely available, and if the results are elevated, they can prompt a doctor, including sports medicine doctor, to assess vitamin D concentration in such patients.

In the group of football players in our study who had vitamin D concentration ≥20 ng/mL, the number of eosinophil cells was significantly higher than in the group with vitamin D deficiency. Moreover, we showed a statistically significant correlation between 25(OH)D levels and the number of eosinophils in peripheral blood. Eosinophils are considered the peripheral effector of the type 2 T helper cell (Th2) arm of immunity involved in the allergic response [67]. However, the correlation between serum vitamin D and eosinophils has not been definitively confirmed. Several studies have shown that lower levels of vitamin D are associated with increased blood eosinophils [31,32]. However, most other studies have not shown any statistical significance of such an association [68,69].

At this stage of the study, we are unable to explain the positive correlation of vitamin D with eosinophil count; this certainly requires further research. In conclusion, our findings regarding the association of vitamin D deficiency with hemogram changes do not support the observations of other authors. Nevertheless, we emphasize that our study is a pilot study.

## 5. Limitations

Our study undoubtedly had some limitations, primarily due to the small sample size, but the group was homogeneous with respect to race, age, physical activity level, exercise load used, body fat, and lifestyle—including diet, sunlight exposure time, and place of residence (Wrocław, Poland). At the same time, we observed a high proportion of athletes with vitamin D deficiency, which obliges us to continue similar studies in a group with optimal vitamin D concentration. In our study, we assessed vitamin D status only during the winter period; to characterize it better, it will be necessary to check 25(OH)D levels also during the summer season. In our research, we limited ourselves to measuring only three cytokines and typical blood count, but it should be emphasized that this was a pilot study.

## 6. Conclusions

Based on the results of our study, we concluded that football players from Poland are not protected against vitamin D insufficiency in winter months. Moreover, Vitamin D deficiency may be associated with an increased pro-inflammatory risk in competitive athletes, which can play an important role in decreased immunity, increase the risk of injuries, or contribute to the onset of overtraining syndrome.

## Figures and Tables

**Table 1 ijerph-18-07124-t001:** Participants’ characteristics.

Characteristics	Mean ± SD (*n* = 41)
Age [years]	22.7 ± 5.3
Body weight [kg]	76.3 ± 7.4
Height [m]	1.82 ± 6.7
Body fat [%]	19.1 ± 3.2
VO_2max_ [mL/kg/min]	55.8 ± 4.0
Career duration [years]	14.7 ± 4.5

VO_2_max: maximal oxygen uptake.

**Table 2 ijerph-18-07124-t002:** Hemogram parameters of the study group.

	Mean ± SD (*n* = 41)
WBC (tys/µL) [4.0–10.0]	5.68 ± 1.62
% NEUTH (%) [40–74]	48.3 ± 9.78
% LYMPH (%) [19–48]	37.4 ± 8.44
% EOS (%) [0.5–6.0]	3.4 ± 2.49
# NEUTH (tys/µL) [1.9–8.0]	2.82 ± 1.39
# LYMPH (tys/µL) [0.9–4.5]	2.06 ± 0.58
# EOS (tys/µL) [0.05–0.5]	0.19 ± 0.14
PLT (tys/µL) [130–440]	219.49 ± 46.29
MPV (fL) [7.2–12]	8.03 ± 0.84
PLR	34.1 ± 5.2
NLR	1.53 ± 1.23

**Table 3 ijerph-18-07124-t003:** Cytokine levels and hemogram parameters (mean ± SD) relative to 25(OH)D levels.

Serum 25(OH)D Concentration (ng/mL)			
	<20 *n* = 32	≥20 *n* = 9	*p*
IL-1β (pg/mL)	2.4 ± 2.2	1.7 ± 1.5	0.572
IL-6 (pg/mL)	0.86 ± 0.78	0.73 ± 0.37	0.613
TNF-α (pg/mL)	1.46 ± 0.26	1.46 ± 0.29	0.971
WBC (tys/µL)	5.64 ± 1.73	5.81 ± 1.29	0.601
% NEUTH (%)	47.5 ± 10.6	51.4 ± 6.2	0.124
% LYMPH (%)	38.4 ± 9.1 *	33.5 ± 4.4 *	0.02
% EOS (%)	3.02 ± 2.07	4.83 ± 3.55	0.060
# NEUTH (tys/µL)	2.77 ± 1.52	3.00 ± 0.83	0.215
# LYMPH (tys/µL)	2.09 ± 0.61	1.94 ± 0.50	0.516
# EOS (tys/µL)	0.168 ± 0.116 *	0.288 ± 0.184 *	0.028
PLT (tys/µL)	219 ± 47	221 ± 51	0.902
MPV (fL)	8.03 ± 0.83	8.00 ± 0.97	0.914
PLR	112.5 ± 35.4	118.2 ± 30.4	0.521
NLR	1.51 ± 0.37 *	1.59 ± 0.40 *	0.035

* *p* < 0.05.

**Table 4 ijerph-18-07124-t004:** Correlation between 25(OH)D levels and IL-6, IL-1β, and TNF-α in athletes.

25(OH)D (ng/mL) (*n* = 41)		
	Rho	*p*
IL-1β (pg/mL)	−0.16	0.34
IL-6 (pg/mL)	−0.14	0.41
TNF0α (pg/mL)	−0.47 *	0.004

* *p* < 0.01.

**Table 5 ijerph-18-07124-t005:** Correlations between 25(OH)D concentration and hemogram parameters.

25(OH)D (ng/mL)		
	Rho	*p*
WBC (tys/µL)	0.08	0.62
% NEUTH (%)	0.49 *	0.02
% LYMPH (%)	−0.50 *	0.02
% EOS (%)	0.26	0.09
# NEUTH (tys/µL)	0.49 *	0.02
# LYMPH (tys/µL)	−0.06	0.71
# EOS (tys/µL)	0.31 *	0.049
PLT (tys/µL)	0.14	0.37
MPV (fL)	0.08	0.63
PLR	0.079	0.612
NLR	0.197	0.204

* *p* < 0.01.

## Data Availability

Data are stored in a de-identified state and can be made available by reasonable and appropriate request.

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
