# Peer review of "Relationships between Vitamin D and Selected Cytokines and Hemogram Parameters in Professional Football Players—Pilot Study"

_ijerph, 2021, doi:10.3390/ijerph18137124_

Round 1
Reviewer 1 Report
Authors aims at investigating the relationship between 25(OH)D concentration and selected cytokines, hemogram parameters in high-rank football players. Authors found that football players from Poland are not protected against vitamin D insufficiency in winter months; and Vitamin D deficiency may be associated with an increased pro-inflammatory risk in well-trained athletes.
This is a nice article, well written, concise. Methodology is robust. Statistical analysis is well conducted.
Some minor issues:
- I would suggest commenting on the relationship between vitamin D levels and cardiac/vascular structure and the potential interplay between VitD/cytokines/cardiovascular structures (J Clin Endocrinol Metab. 2012 Oct;97(10):3717-23.; J Intern Med. 2013 Mar;273(3):253-62.).
- Can you correlate vitD/cytokines data with any performance indexes?
- Can you specify whether the cutoff of 20ng/ml for Vit D has been used?
Reviewer 2 Report
Dear Dr. KsiÄ…ĹĽek and Dr. SĹ‚owiĹ„sk –Lisowska
Thanks for your manuscript on vitamin D and cytokines in professional football players.
First, please make a table to summarize the characters of the participants. (Line 80-82).
Now, let us go over the Newcastle-Ottawa Scale (NOS), which is used for assessing the quality of pilot studies.
|
Cohort Star Template |
Selection of cohorts |
Comparability of cohorts |
Outcome |
Rank |
|||||
|
Study |
Representativeness |
Selection of the |
Ascertainment |
Demonstration that |
Comparability of cohorts on |
Assessment |
Was follow up |
Adequacy of |
Thresholds for converting the Newcastle-Ottawa scales to AHRQ standards (good, fair, and |
|
Before modification |
* |
NA |
* |
NA |
* |
* |
NA |
NA |
Poor quality |
|
After modification |
* |
NA |
* |
* |
* |
* |
* |
NA |
good quality or at less fair quality |
Please answer the question below,
- To get the star for “ Demonstration that outcome of interest was not present at the start of study”, did you measure the baseline of the vitamin D and cytokines and hemogram factors you observed in another season as a baseline ? Just to confirm all the participants are not vitamin D insufficient or deficiency at the beginning of the enrollment. If not, please cite some data from other studies in the summer or just write it as a limitation.
- This is for the star about “ Was follow up long enough for outcomes to occur”. As you mentioned, “our study were evaluated in December and January “. ( Line 175) Could you mention December and January in the “ blood test” paragraph ( Line 100)? Could you mention a little bit why you choose this time for the discussion part? Please give evidence whether is it long enough for vitamin D (insufficient or deficiency happen?)
Thank you so much again and I am looking for your answers.
Round 2
Reviewer 2 Report
I accepted the present form. Best of luck in your future endeavors!